# Screening and Characterization of Shark-Derived VNARs against SARS-CoV-2 Spike RBD Protein

**DOI:** 10.3390/ijms231810904

**Published:** 2022-09-18

**Authors:** Yu-Lei Chen, Jin-Jin Lin, Huan Ma, Ning Zhong, Xin-Xin Xie, Yunru Yang, Peiyi Zheng, Ling-Jing Zhang, Tengchuan Jin, Min-Jie Cao

**Affiliations:** 1College of Ocean Food and Biological Engineering, Jimei University, Xiamen 361021, China; 2CAS Key Laboratory of Innate Immunity and Chronic Disease, Division of Life Sciences and Medicine, University of Science & Technology of China, Hefei 230007, China

**Keywords:** SARS-CoV-2, RBD, variants, shark, VNAR, bi-paratopic VNAR

## Abstract

The receptor-binding domain (RBD) of the SARS-CoV-2 spike protein is the major target for antibody therapeutics. Shark-derived variable domains of new antigen receptors (VNARs) are the smallest antibody fragments with flexible paratopes that can recognize protein motifs inaccessible to classical antibodies. This study reported four VNARs binders (JM-2, JM-5, JM-17, and JM-18) isolated from *Chiloscyllium plagiosum* immunized with SARS-CoV-2 RBD. Biolayer interferometry showed that the VNARs bound to the RBD with an affinity K_D_ ranging from 38.5 to 2720 nM, and their Fc fusions had over ten times improved affinity. Gel filtration chromatography revealed that JM-2-Fc, JM-5-Fc, and JM-18-Fc could form stable complexes with RBD in solution. In addition, five bi-paratopic VNARs, named JM-2-5, JM-2-17, JM-2-18, JM-5-18, and JM-17-18, were constructed by fusing two VNARs targeting distinct RBD epitopes based on epitope grouping results. All these bi-paratopic VNARs except for JM-5-18 showed higher RBD binding affinities than its component VNARs, and their Fc fusions exhibited further enhanced binding affinities, with JM-2-5-Fc, JM-2-17-Fc, JM-2-18-Fc, and JM-5-18-Fc having K_D_ values lower than 1 pM. Among these Fc fusions of bi-paratopic VNARs, JM-2-5-Fc, JM-2-17-Fc, and JM-2-18-Fc could block the angiotensin-converting enzyme 2 (ACE2) binding to the RBD of SARS-CoV-2 wildtype, Delta, Omicron, and SARS-CoV, with inhibition rates of 48.9~84.3%. Therefore, these high-affinity VNAR binders showed promise as detectors and therapeutics of COVID-19.

## 1. Introduction

Coronavirus disease 2019 (COVID-19), caused by severe acute respiratory syndrome coronavirus 2 (SARS-CoV-2), has infected over 607 million people worldwide, with approximately 6.49 million reported deaths as of 2 September 2022. The spike glycoprotein homotrimer (S) on the SARS-CoV-2, as with other coronaviruses, is critical for receptor binding and viral entry. It contains two functional subunits, S1 and S2. The S1 subunit facilitates the binding of the host cell receptor via the interaction between its C-terminal receptor-binding domain (RBD) and human angiotensin-converting enzyme 2 (ACE2), whereas the S2 subunit catalyzes fusion of the viral and host cell membranes [1]. Surprisingly, the binding affinity of SARS-CoV-2 S1 to ACE2 is higher than that of SARS-CoV S1 [2]. SARS-CoV-2 RBD can induce neutralizing antibodies in animals and is necessary for virus infection in host cells. Therefore, it can act as a good target for developing vaccines and neutralizing antibodies.

The number of COVID-19 infections is still increasing due to the spread of SARS-CoV-2 variants. Many countries have approved several COVID-19 vaccines for emergency use, including mRNA vaccines, viral vector vaccines, inactivated whole-virus SARS-CoV-2 vaccines, and protein-based vaccines [3,4]. Appearances of SARS-CoV-2 variants, especially the Delta variant and Omicron variant, have created challenges for the ongoing vaccination drive against COVID-19. Studies have extensively characterized the antibody against SARS-CoV-2 in response to infection [5,6,7,8,9,10,11] and vaccination [12,13,14,15,16]. Antibody analysis reveals that recovering COVID-19 patients generate potent neutralizing antibodies against the SARS-CoV-2 S protein. The existing monoclonal antibodies (mAbs) in the plasma of convalescent patients have been used as therapeutic alternatives for COVID-19 [17,18]. Moreover, various mAbs that can neutralize SARS-CoV-2 have been isolated from patients [19,20,21,22,23,24,25]. Five neutralizing mAbs, including Casirivimab/Imdevimab (developed by Regeneron in 2020, revoked), Sotrovimab (developed by GlaxoSmithKline in 2021, revoked), Bamlanivimab/Etesevimab (developed by Lilly in 2021, revoked), Cilgavimab/Tixagevimab (developed by AstraZeneca in 2021), and Bebtelovimab (developed by Lilly in 2022), have been approved by U.S. Food and as drug administration for emergency use.

The emerging novel variants are due to natural selection as they can infect/reinfect the human population and escape the immune response. Emergency use authorizations for three neutralizing mAbs were revoked due to ineffectiveness for these variants. Vaccination and passive immunization with Ig-based antibodies, as selection pressure, may promote the emergence of new variants, such as Omicron variants, which evade a broad spectrum of neutralizing antibodies, leading to decreased protection. In addition to the conventional mAbs, heavy-chain-only antibodies (HCAbs) from camelids [26] and Ig new antigen receptors (IgNARs) from sharks and other cartilaginous fish [27], consisting of two identical heavy chains devoid of light chains, are alternatives for COVID-19 treatment. IgNARs contain one variable domain (VNAR) and five constant (C) domains, while VNAR has four highly conserved framework regions (FRs), two hypervariable loops (HV), and two highly variable complementary determining regions (CDRs). VNAR is the smallest natural antibody in the animal kingdom (~12 kDa). Compared to the conventional antibody, VNAR has various advantages, including high affinity and specificity, small size, improved thermo-stability, low immunogenicity, and excellent tissue penetration characteristics. Moreover, it can be engineered into multi-valent and multi-specific antigen-binding formats widely applied in oncotherapy, diagnosis, monitoring of disease, and prevention of viral infection [28,29,30]. A VNAR binder targeting the Hepatitis B virus pre-core protein with an affinity K_D_ of 53 nM was isolated from a semi-synthetic wobbegong shark library. The intracellular VNAR could disrupt the processing of the viral pre-core antigen and reduce HBeAg secretion [31]. Goodchild and colleagues generated a VNAR library from a nurse shark immunized with inactivated Zaire ebolavirus, and the isolated VNARs showed specificity for viral nucleoprotein and exhibited cross-reactivity for multiple Ebolavirus species that cause disease in humans [32].

In this study, VNAR phage libraries from *Chiloscyllium plagiosum* immunized with SARS-CoV-2 RBD were generated, and four VNAR binders targeting SARS-CoV-2 RBD (JM-2, JM-5, JM-17, and JM-18) were isolated. Bi-paratopic VNARs binding with non-overlapping epitopes were constructed based on the epitope binning results. The monomer VNARs and bi-paratopic VNARs were biochemically characterized, and results showed that some of them could effectively block the interaction between the RBD protein of SARS-CoV-2 and its variants with the ACE2 receptor. These results illustrate that these VNARs offer a basis for the future development of anti-SARS-CoV-2 antibodies.

## 2. Results

### 2.1. Four Unique VNAR Binders Targeting SARS-CoV-2 RBD Were Identified from Immunized C. plagiosum

Three *C. plagiosum* were immunized with recombinant SARS-CoV-2 RBD of high purity to obtain SARS-CoV-2 neutralizing antibodies with high affinity and specificity. The immunization consisted of three subcutaneous injections and two intravenous tail injections. Peripheral blood mononuclear cells were isolated from the immunized *C. plagiosum*. Reverse transcription was performed using the extracted RNA as a template. The VNAR coding region (about 340 bp) was amplified using PCR (Figure 1A). The PCR fragments and phagemid vector pR2 were digested, ligated, and transformed into TG1 cells. Finally, a VNAR phage library containing 3.0 × 10^8^ transformants was successfully constructed. The diversity of the VNAR phage library was determined to be 88%. Additionally, the insertion rate of VNAR genes (95.8%) was evaluated using PCR on 24 randomly picked clones (Figure 1A). Bio-panning was performed thrice to enrich RBD-binding clones. An individual phage was randomly picked, and their RBD-binding activity was evaluated via monoclonal phage ELISA. A total of 84 from 95 clones were identified as positive clones, as they had a higher binding rate than the negative control (Figure 1B). Most ELISA-positive colonies showed high binding activity to the RBD. The positive colonies were sequenced, and the repeated sequences were removed. Four unique VNARs (JM-2, JM-5, JM-17, and JM-18) with distinct CDR3 were obtained based on the amino acid sequencing and alignment (Figure 1C).

The identified VNARs were expressed with a mammalian expression vector in HEK293F cells. The C terminus of the identified VNARs was fused to a TEV protease cleavage site and a human IgG1 Fc in a mammalian expression vector to configure the VNAR into an IgG-like molecule. The VNAR-Fc fusion chimeric antibodies were purified from the culture supernatant using an rProtein A column. VNAR monomers without Fc fragments were prepared as follows: the VNAR-Fc fusions were digested with the TEV enzyme (6× His tagged) and passed through rProtein A and Ni-NTA column. SDS-PAGE analysis showed that VNAR-Fc fusions (JM-2-Fc, JM-5-Fc, JM-17-Fc, and JM-18-Fc) and Fc-free VNARs (JM-2, JM-5, JM-17, and JM-18) were highly purified (Figure 1D).

### 2.2. Characterization of RBD-Targeted VNARs

To investigate the thermal stability of the isolated VNARs, we measured the melting temperatures (T_m_) of VNARs via a thermal shift assay. The results showed that the T_m_ values were 56.39 °C, 54.92 °C, 54.38 °C, and 55.62 °C for JM-2, JM-5, JM-17, and JM-18, respectively, indicating that the four RBD-targeted VNARs are highly thermostable.

We first used gel filtration chromatography to preliminarily verify the binding abilities of the VNAR-Fc fusions to SARS-CoV-2 RBD, and found that JM-2-Fc, JM-5-Fc, JM-17-Fc, and JM-18-Fc all formed a complex with the RBD in solution, despite less complex formed by JM-2-Fc and JM-17-Fc (Figure 2A). Biolayer interferometry (BLI) was further performed to determine the RBD binding affinity of the VNARs and showed that JM-5 and JM-18 bound the RBD with K_D_ values of 38.5 nM and 60.3 nM, while JM-2 and JM-17 bound the RBD with K_D_ values of 429 nM and 2720 nM, respectively (Figure 2B and Table 1). These BLI results are in agreement with the gel filtration results that JM-5 and JM-18 bind RBD more tightly than JM-2 and JM-17. We also measure the RBD binding affinity of the VNAR-Fc fusions and found that VNAR-Fc fusions exhibited highly enhanced RBD binding affinities over VNAR monomers, with K_D_ values of 28.3 nM, 3.88 nM, 211 nM, and 9.20 nM for JM-2-Fc, JM-5-Fc, JM-17-Fc, and JM-18-Fc, respectively (Figure 2C and Table 1). Finally, ELISA was performed to characterize the RBD binding of these VNAR-Fc fusions. Consistent with gel filtration and BLI results, ELISA showed that JM-5-Fc and JM-18-Fc fusions had stronger RBD binding abilities than JM-2-Fc and JM-17-Fc fusions (Figure 3A). The 50% maximal effective concentration (EC_50_) values for JM-5-Fc and JM-18-Fc binding to the RBD were 0.190 nM and 1.437 nM, respectively, which were even lower than the EC_50_ value for human ACE2-Fc recombinant protein (4.146 nM).

We next tested the binding of our VNARs to the RBD of two major SARS-CoV-2 variants, Delta and Omicron. ELISA showed that all four VANR-Fc fusions retained binding to the Delta RBD and were even more active to delta than to WT RBD (Figure 3B). However, only JM-5-Fc retained a strong binding activity to Omicron RBD (EC_50_ value of 0.355 nM), whereas the binding of the other three VANR-Fc fusions to the Omicron RBD was abolished or significantly reduced (Figure 3C). Furthermore, we also tested the binding of our VNARs to SARS-CoV RBD and showed that all four VNAR-Fc fusions reacted with SARS-CoV RBD, and JM-5-Fc bound best, with an EC_50_ value of 0.791 nM (Figure 3D). Overall, JM-5-Fc is stronger than the other three VNARs in binding the RBDs of WT, Delta, and Omicron, and across-reacts with SARS-CoV RBD with high activity.

### 2.3. RBD-ACE2 Blockage of VNARs

SARS-CoV-2 infection is initiated by the binding of RBD to the cell surface of ACE2. To assess the ability of VNAR-Fc fusions in blocking RBD-ACE2 interaction, a BLI assay was conducted. For blocking ACE2-WT RBD, JM-2-Fc was best, followed by JM-18-Fc, and JM-5-Fc and JM-17-Fc were weakest (Figure 4A). A similar situation was observed for blocking ACE2-Delta RBD, and possibly because the four VNAR-Fc fusions bound more strongly to Delta RBD than to WT RBD, they have a higher blocking activity against ACE2-Delta RBD than against ACE2-WT RBD (Figure 4B). Maybe due to the reduced binding activity, none of the VNAR-Fc fusions were obviously active in blocking ACE2-Omicrn RBD interaction (Figure 4C). JM-2-Fc and JM-5-Fc were also active in blocking ACE2-SARS-CoV RBD interaction (Figure 4D). Among the four VNARs, JM-2-Fc was most effective in blocking the ACE2 binding to the WT, Delta, and SARS-CoV RBDs, with inhibition rates of 73.2%, 86.6%, and 52.5%, respectively (Figure 4A,B,D).

### 2.4. Epitope Competition of VNARs

BLI was performed to analyze the competition between the isolated VNARs for RBD binding. Biosensors loaded with biotinylated WT SARS-CoV-2 RBD were bound with the first VNAR-Fc fusion to reach binding saturation, and then with a second VNAR-Fc fusion with the same concentration (500 nM). A signal increase in the binding curve indicated a noncompetitive relationship between the two VNARs. We found that competition was only observed between JM-5 and JM-17, while the other VNARs did not compete with each other (Figure 5A,B), indicating that our four VNARs target three independent RBD epitopes.

To further explore the epitope information of our VNARs, we measured the epitope competition of our VNARs against the previously reported alpaca-derived variable domain of heavy-chain antibodies (VHHs) (aRBD-2, aRBD-5, and aRBD-7) [33] whose epitopes are located on the receptor-binding motif (RBM). The RBD-coated biosensors were sequentially bound to VNAR-Fc fusions and alpaca-derived VHH-Fc fusions with the same concentration (500 nM). The binding signal shifting value with ~0.1 in the curve indicates the presence of a competitive relationship between the two nanobodies. We only observed competition between JM-18 and aRBD-2 (Figure 6A,B), suggesting that JM-18 may bind an epitope on the RBM. In contrast, JM-2, JM-5, and JM-17 have no competition with aRBD-2, aRBD-5, and aRBD-7, suggesting that they possibly bind to the epitopes on the RBD core.

### 2.5. Prediction of the Binding Sites of VNARs on RBD Using Docking Simulation

Based on the epitope competition results, docking simulation was conducted to predict and compare the binding affinity of our VNARs with WT SARS-CoV-2 RBD. The structural models of the complex between VNARs and the RBD were constructed using the modeling program ClusPro. The ΔG values of VNARs’ binding for the RBD were −4.3, −8.3, −2.0, and −3.7 kcal·mol^−1^ for JM-2, JM-5, JM-17, and JM-18, respectively, which are in line with the K_D_ values of these VNARs. The binding of VNARs to the RBD was displayed by PyMOL (Figure 7). JM-2, JM-5, JM-17, and JM-18 formed 17 hydrogen bonds and two salt bridges, five hydrogen bonds and two salt bridges, ten hydrogen bonds and three salt bridges, and 18 hydrogen bonds and four salt bridges with the RBD, respectively (Figure 8).

### 2.6. Construction and Characterization of High-Affinity Bi-Paratopic Antibodies

Based on epitope competition results, five bi-paratopic VNAR constructs (JM-2-5, JM-2-17, JM-2-18, JM-5-18, and JM-17-18) were designed by connecting two VNAR sequences through a (GGGGS)_3_ flexible linker (Figure 9A). The constructs were also expressed with HEK293F cells and purified (Figure 9B) using rProtein A. The binding affinities of the constructs for the WT SARS-CoV-2 RBD were also studied using BLI. Compared to monovalent VNARs, the constructs showed an enhanced binding affinity (except for JM-5-18), with K_D_ values of 6.39 nM, 32.1 nM, 30.7 nM, and 35.7 nM for JM-2-5, JM-2-17, JM-2-18, and JM-17-18, respectively (Figure 9C and Table 1). After fusing to Fc, their binding affinities were further improved, with K_D_ values even lower than 1 pM (Figure 9D and Table 1).

Consistent with the BLI results, ELISA showed that the five bi-paratopic VNAR-Fc fusions had at least 10-fold stronger activity than their monomers in binding WT SARS-CoV-2 RBD, with EC_50_ values in the sub-nM range (Figure 10A). In addition, the binding of the five bi-paratopic VNAR-Fc fusions to the RBDs of Delta, Omicron variants, and SARS-CoV was also highly improved (Figure 10B–D).

The blocking ability of bi-paratopic VNAR-Fc fusions against RBD-ACE2 interaction was further determined. JM-2-5-Fc, JM-2-17-Fc, and JM-2-18-Fc fusions exhibited similar activities in blocking ACE2 binding to the RBD of WT, Delta, Omicron, or SARS-CoV. The inhibition rates of these three bi-paratopic VNARs were ~50%, ~70%, ~60%, and ~80% for the RBD of SARS-CoV-2 WT, Delta, Omicron, and SARS-CoV, respectively (Figure 11). Importantly, bi-paratopic VNAR-Fc fusions acquired blocking activities against the interaction between Omicron RBD and ACE2 (Figure 11C).

## 3. Discussion

SARS-CoV-2 is the culprit that causes COVID-19, and its global spread poses a serious threat to human health. As a result, the development of vaccines, monoclonal antibodies, and small-molecule direct-acting antiviral medications for COVID-19 is necessary. SARS-CoV-2 infects epithelial cells through the interaction between RBD and ACE2 [34]. Therefore, RBD-targeting antibodies are promising as prophylactics and therapeutics for COVID-19. As with many viruses, SARS-CoV-2 launches its cellular invasion through its heavily glycosylated S protein. A total of 22 highly occupied N-linked glycosylation sites have been identified in the S protein, with two N-linked glycosylation sites (N331 and N343) in RBD [35]. A further study showed that N-glycosylation of the RBD is not only critical for viral internalization but also shields the virus from antibody neutralization [36]. In this study, the SARS-CoV-2 RBD protein used to immunize sharks was prepared using a mammalian cell expression system; N331 and N343 of this RBD protein should, thus, be glycosylated, as performed in the natural virus. Therefore, the RBD-targeting VNARs isolated here should retain binding to the native RBD present on authentic SARS-CoV-2 without being affected by glycosylation.

Most developed SARS-CoV-2 neutralizing mAbs were escaped from by the SARS-CoV-2 Omicron variant, which carries numerous mutations in the RBD protein [37]. VHHs from camelids or VNARs from sharks offer unique binding capabilities to the RBD protein due to their smaller size, especially in the regions that are not readily susceptible to conventional mAbs, making them excellent alternatives to conventional antibodies. Thus far, a large number of SARS-CoV-2-neutralizing VHHs have been reported [4,5,6,7,8,9,10,11,12,13,14,15,16,17,18,19,20,21,22,23,24,25,26,27,28,29,30,31,32,33,33,34,35,36,37,38,39,40,41,42,43,44,45,46]. However, SARS-CoV-2-neutralizing VNARs are still rarely reported [47,48]. In this study, we characterized four unique VNARs isolated from RBD-immunized *C. plagiosum*. Amino acid sequence alignment showed that these VNARs have distinct CDR3 sequences. Among the four VNARs, JM-5 and JM-18 bound to the RBD with K_D_ values of 38.5 and 60.3 nM, respectively, which are comparable to the affinity K_D_ values of previously reported RBD-targeting VNARs (K_D_ values, 17.2–63.0 nM) [48]. After fusing with the IgG1 Fc fragment to form JM-5-Fc and JM-18-Fc, their RBD-binding affinity increased by ~10 times, with K_D_ values of 3.88 nM and 9.20 nM, respectively, even higher than the affinity of some mAbs isolated from lymphocytes of convalescent COVID-19 patients [24,49,50]. This was attributed to the bivalent nature of dimerized VNAR-Fc fusion antibodies, with a similar trend in affinity to previously reported nanobodies [42,51].

ELISA showed that JM-5-Fc and JM-18-Fc bind WT RBD with EC_50_ values in the nanomolar range (Figure 3A), similar to some previously reported VNARs [47]. A previous study revealed that the shark-derived 20G6 antibody can effectively bind to WT RBD, but it loses the binding ability to the Omicron variant RBD. This was due to the disruption of the structure of the β-strand on the Omicron RBD by S375F mutation, thus impairing the binding with 20G6 [52]. In this study, VNARs were screened against RBD derived from the ancestral SARS-CoV-2, and their performance against the newly detected variants of the virus and SARS-CoV was explored. The mutations in Delta RBD improved the binding and blocking abilities of the four VNAR-Fc fusions, according to our findings. However, their activities against Omicron RBD were significantly impaired. This might be due to the large number of mutation sites in Omicron RBD. Even though mutations in RBD help the virus escape from the host immunity, the JM-5-Fc antibody remained a potent binder to Omicron RBD (EC_50_ = 0.355 nM). In addition, JM-5-Fc could effectively bind with SARS-CoV RBD (EC_50_ = 0.791 nM), indicating that JM-5-Fc may be a broad-spectrum antibody for sarbecovirus.

VNARs can effectively access the recessed epitope due to the protruding CDR3 sequence, thus underscoring the utility of neutralizing VNAR. Herein, epitope binning identified three nonoverlapping epitope bins in the RBD domain recognized by VNAR-Fc fusions. Moreover, JM-5-Fc and JM-17-Fc competed for the overlapping epitopes, while JM-2-Fc and JM-18-Fc occupied separate epitope bins. The crystal structure showed that the aRBD-2 nanobody from alpaca recognizes the lateral loop of the RBM of RBD, while aRBD-5 and aRBD-7 from alpaca bind to the concave surface anchored by the β-hairpin of the RBM [33]; these three alpaca nanobodies collectively occupy almost the entire RBM surface. In this study, JM-2, JM-5, and JM-17 have no competition with these three alpaca nanobodies and cross-react with the SARS-CoV RBD, and it can be concluded that JM-2, JM-5, and JM-17 should bind to epitopes on the RBD core. The exception is JM-18, which competes with alpaca-derived aRBD-2, but also has low cross-reactivity with the SARS-CoV RBD, indicating that JM-18 may target a RBD core epitope close to the RBD epitope of aRBD-2 but far away from the epitope of the other three VNARs. Nonetheless, the specific binding epitopes of these VNARs need to be finalized by structural biology in the future.

On the basis of the epitope grouping results, we constructed bi-paratopic VNARs by fusing VNARs targeting independent RBD epitopes. This is the first study assessing bi-paratopic VNARs targeting SARS-CoV-2. Previous studies have demonstrated that multi-valent nanobodies formed by tandem fusing have a stronger binding affinity than the monomer nanobody [46,53], even if one component nanobody lost observable binding affinity to some SARS-CoV-2 variants; the overall affinity was effectively improved when transformed into bi-paratopic form [33]. Herein, five bi-paratopic VNARs with significantly increased RBD-binding abilities were generated. Especially, the bi-paratopic JM-2-5 has at least a 6-fold higher RBD binding affinity than their component monomers. The bi-paratopic VNAR-Fc fusions showed ultra-high binding abilities to the WT RBD with K_D_ values even lower than 1 pM. The bi-paratopic VNAR-Fc fusions also showed enhanced binding ability to the RBDs of the SARS-CoV-2 variant and SARS-CoV, with EC_50_ values in sub-nanomolar to picomolar RBDs (Figure 10). Among these bi-paratopic VNAR-Fc fusions, JM-2-5-Fc, JM-2-17-Fc, and JM-2-18-Fc showed potent activities in blocking ACE2 binding to the RBDs of SARS-CoV-2 variants and SARS-CoV, which predicts the broad-spectrum neutralizing activity of these bi-paratopic VNAR-Fc fusions against the emerging SARS-CoV-2 variants and SARS-CoV.

## 4. Materials and Methods

### 4.1. Plasmids and Cell Culture

An engineered pTT5 plasmid with a TEV enzyme cleavage site, a human IgG1 Fc at the C terminus, and an IFNA1 signal peptide at the N terminus were used for VNAR-Fc fusion expression. Electro-competent *Escherichia coli* TG1 cells were preserved in our laboratory.

Human embryonic kidney (HEK) 293F cells were obtained from TJ-Lab. The cells were maintained in Union-293 (Union, Shanghai, China) supplemented with 100 units/mL of penicillin-streptomycin (Gibco, Carlsbad, CA, USA), then cultured at 5% CO_2_ and 37 °C. DMEM (VivaCell, Denzlingen, Germany) and polyethyleneimine reagent (PEI, Polyscience, Warrington, PA, USA) were used for cell transfection.

### 4.2. Protein Expression and Purification

SARS-CoV-2 RBD (amino acids [aa]; 321–591), the human ACE2 extracellular domain (aa; 19–615), and the identified VNARs and bi-paratopic VNARs were constructed into a pTT5-TEV-Fc vector and prepared as previously reported [54,55]. Bi-paratopic VNAR sequences were designed by connecting two VNAR sequences through a (GGGGS)_3_ flexible linker. Plasmids were propagated in *E. coli* (2× TY medium supplemented with 100 μg/mL of ampicillin) at 37 °C overnight. The recombinant plasmids were transiently transfected into HEK293F cells for three days. The cell culture supernatants were then obtained via centrifugation at 5000 g for 10 min. The supernatants were diluted (1:1) with running buffer (20 mM Na_2_HPO_4_ and 150 mM NaCl (pH 7.0)), filtered through a 0.22 μm filter before purification, and loaded on a rProtein A column (Cytiva, Marlborough, MA, USA). The bound protein was eluted with 100 mM acetic acid on a Unique Autopure 25 (Inscinstech, Suzhou, China). The purified fusion proteins were digested with the 6× His-tagged TEV enzyme to remove the IgG1 Fc fragment. The undigested fusion protein, Fc fragment, and the TEV enzyme were sequentially removed using rProtein A and a Ni-nitrilotriacetic acid (NTA) column. Fc-free recombinant VNAR proteins were collected from the flow-through. SDS-PAGE was used to analyze the expression and purity of recombinant proteins.

### 4.3. Biotinylation of RBD and ACE2-Fc

The Biotinylation Kit (Genemore, Suzhou, Jiangsu, China) was used for biotinylation of RBD and ACE2-Fc following the manufacturer’s protocol. Briefly, the protein was dissolved in PBST (PBS (pH 7.4) supplemented with 0.02% Tween 20) with a concentration greater than 2 mg/mL and incubated with biotin in the dark at room temperature for 1 h. The solution was subsequently loaded onto the PD MiniTrap™ G-25 Desalting Column (Cytiva) pre-equilibrated with PBST, then eluted with PBST.

### 4.4. Phage Library Construction

RBD was used as an antigen for *C. plagiosum* (obtained from Xiamen, China) immunization. The immunized phage library was generated as described by Ma et al. with some modifications [42]. Briefly, the three *C. plagiosum* were immunized thrice with 250 μg of RBD emulsified with an equal volume of Freund adjuvant (Sigma-Aldrich, St. Louis, MO, USA) via subcutaneous injection at intervals of 10 days. The three *C. plagiosum* were subsequently immunized twice with 250 μg of RBD in phosphate-buffered saline (PBS) via tail vein injection at intervals of 30 days. More than 1 × 10^7^ lymphocytes were isolated from peripheral blood after 15 days of the final boost. Total RNA from the lymphocytes was isolated using a Total RNA kit (Omega Bio-Tek, Norsross, GA, USA) following the manufacturer’s protocol. First-strand cDNA was synthesized from 4 μg of total RNA per reaction using a PrimeScript^TM^ II first-strand cDNA synthesis kit (TaKaRa, Dalian, China) following the manufacturer’s protocol. PCR was used to amplify VNAR using the primers below: forward primer: GCTGCACAGCCTGCTATGGCAACTCAACGGGTTGAACAAACACCGAC; reverse primer: GAGTTTTTGTTCGGCTGCTGCTGGTTTTACAGTCAGAATGGTGCCGC. The pR2 phagemid was amplified using the following primers: forward primer: AGCAGCCGAACAAAAACTCATCTCAGAAGAG; reverse primer: CCATAGCAGGCTGTGCAGCATAGAAAGGTACCACTAAAGGAATTGC. It was then digested with the Nde I restriction enzyme (New England Biolabs, Ipswich, MA, USA) to destruct the template phagemid. The VNAR fragments (2 pmol) and 0.5 pmol of the amplified pR2 vector were mixed and diluted to 50 μL. An equal volume of Uniclone Seamless Cloning Mix (2×) (Genesand Biotech, Beijing, China) was added to the mixture, then incubated at 50 °C for 1 h. A Cycle-Pure kit (Omega Bio-Tek) was used to purify the ligation product. The purified product was used to transform the freshly prepared TG1 cells via the BTX ECM 399 electroporation system (Harvard Apparatus, Holliston, CA, USA) with the following settings: 2.5 kV and 5 ms. The transformed cells were re-suspended with 200 μL of 2× TY culture medium and incubated at 37 °C for 1 h. The transformants were spread on five 150 mm 2× TY agar plates containing 2% glucose and 100 μg/mL of ampicillin, then cultured at 37 °C overnight. The colonies were scraped from the plates, and aliquots of the library stock were flash-frozen and stored at −80 °C. Library size was calculated via serial dilution of aliquots. The positive rate of the constructed library was determined via colony PCR. To determine the diversity of the library, 100 colonies were selected and subjected to DNA sequencing.

### 4.5. Biopanning and Selection of Positive Clones

The phage library (200 μL) was inoculated into 200 mL of 2× TY to amplify the phages. Phage particles with VNAR were rescued from the library using the KM13 helper phage. Biopanning was conducted using phage display technology. RBD was diluted in GFBE (2 mM EDTA, 20 mM Tris, and 250 mM NaCl (pH 8.0)) to a concentration of 100 μg/mL and used to coat Maxisorp Nunc-Immuno plates (Thermo Fisher Scientific, Waltham, MA, USA) at 4 °C overnight. An uncoated well was used as a negative control in parallel with panning. The phages were blocked with MPBS (PBS supplemented with 5% milk powder) at room temperature for 2 h and rinsed with PBS, and then about 1 × 10^11^ PFU of the library phages were added for the selection. The unbound phages were washed 20 times with PBST (PBS supplemented with 0.1% Tween 20), while bound phages were eluted via digestion with 100 μL of trypsin (0.5 mg/mL) at room temperature for 1 h. The eluted phages were used to infect exponentially growing *E. coli* TG1, and then plated on an LB agar plate (100 μg/mL of ampicillin). The bacteria were collected and subjected to a new round of phage amplification for the second and third rounds of panning.

A total of 95 individual clones were randomly selected after the third round of panning and identified using monoclonal phage ELISA. The monoclonal phage was rescued with helper phage KM13 and added to the well coated with 1 μg/mL of RBD, then incubated at room temperature for 1 h. The wells were then washed four times with PBST. The HRP-anti-M13 antibody (SinoBiological, Beijing, China) was added to the well. Each well was washed thrice with PBST, then TMB (Beyotime, Shanghai, China) was added and incubated in the dark at room temperature for 5 min. The reaction was stopped using 50 μL of 1 M sulfuric acid. Infinite M200Pro (Tecan, Männedorf, Switzerland) was used to measure the absorbance at 450 nm. The clones with OD_450_ values higher than 1.0 were defined as positive clones. All positive clones were sequenced and grouped based on their amino acid sequences of complementary determining regions (CDRs).

### 4.6. Determination of Melting Temperatures of VNARs via Thermal Shift Assay

A thermal shift assay was conducted using a QuantStudio6 Flex (Applied Biosystem, Foster City, CA, USA) to measure the melting temperatures (T_m_) of VNARs. Briefly, Fc-free VNARs were mixed with 20 mM HEPES, 150 mM NaCl buffer (pH 7.5), and 5× SYPRO Orange Protein Gel Stain (Sigma-Aldrich) to a final concentration of 0.5 mg/mL. The program conditions were as follows: heating to 25 °C at a ramp rate of 1.6 °C/s, holding at 25 °C for 2 min, and heating to 95 °C at a continuous ramp rate of 0.1 °C/s. GraphPad Prism 5 software (GraphPad, San Diego, CA, USA) was used to calculate the T_m_ values.

### 4.7. Gel Filtration Chromatography

Gel filtration chromatography was used to assess the interaction between RBD and the VNARs in solution. Briefly, RBD, VNARs, and their mixture (1 nmol of RBD mixed with 1 nmol of VNAR-Fc fusions) were run over a Superdex 200 column (Cytiva) at 0.5 mL/min using Unique Autopure 25 (Inscinstech, Suzhou, China).

### 4.8. Affinity Determination

Biolayer interferometry technology (BLI) with a Fortebio’s BLItz (Sartorius AG, Goettingen, Germany) was used for the analysis of binding affinity K_D_ values of VNARs binding to the RBD antigen. All proteins were dissolved in PBST (PBS (pH 7.4) supplemented with 0.02% Tween 20). The Streptavidin (SA) and Protein A biosensors (Pall ForteBio) were hydrated in PBST for 10 min, then equilibrated in PBST for 30 s before loading the protein. The VNARs (500 nM) were coupled to Protein A biosensors (Pall ForteBio) to measure the binding affinity of VNARs. A second equilibration step was performed for 90 s after protein loading. Serial dilutions of the RBD protein were injected over the biosensor for 180 s, followed by 180 s of dissociation. The binding affinity of VNAR-Fc fusions to the RBD antigen was measured as follows: a biotinylated RBD (0.5~1 mg/mL) was loaded on SA biosensors to a load threshold above 1.0 nm, then incubated with a series of diluted VNAR-Fc fusions at room temperature for 240 s, followed by 240 s of dissociation. The binding curves were fit in a 1:1 binding model using BLItz Pro^TM^ software for data acquisition and data analysis. The association (ka) and dissociation rates (kd) were monitored, and the equilibrium dissociation constant (K_D_) was obtained.

### 4.9. Enzyme-Linked Immunosorbent Assay (ELISA)

Maxisorp Nunc-Immuno plates (Thermo Fisher Scientific) were coated with 10 μg/mL of RBD and blocked as previously described. The noncompetitive ELISA of purified VNAR-Fc fusions and the ACE2-Fc binding assay were conducted as follows; VNAR-Fc fusions and ACE2-Fc solutions serially diluted (from 10^4^ to 10^−3^ nM) were added to each well, and incubated at room temperature for 1 h. The wells were washed four times with PBST, then bound VNAR-Fc fusions and ACE2-Fc were detected using a monoclonal anti-IgG Fc-HRP antibody (Abcam, Cambridge, MA, USA).

### 4.10. ACE2-RBD Blocking Assay

The ACE2-RBD blocking assay was performed with a Fortebio’s BLItz (Sartorius AG). Biotinylated RBD was loaded on SA biosensors to a load threshold above 1.0 nm. RBD-coated biosensors were coupled to VNAR-Fc fusions (1 μM) for 480 s after an equilibration step. The VNAR-Fc/RBD-coated biosensors were then coupled to a 3 μM solution of ACE2 for 480 s. VNAR was replaced with PBST and used as a control. The binding curves were fit in a 1:1 binding model using BLItz Pro^TM^ software, version 1.2.1.5 (Sartorius AG) for data acquisition and data analysis.

### 4.11. Epitope Binning

Biotinylated RBD was loaded on SA biosensors for competition-binding assays. RBD-coated biosensors were coupled to one of the VNAR-Fc fusions (500 nM) for 120 s after an equilibration step. The VNAR-Fc/RBD-coated biosensors were then coupled to a 500 nM solution of another VNAR-Fc or VHH-Fc fusion for 120 s. The binding curves were fit in a 1:1 binding model using BLItz Pro^TM^ software for data acquisition and data analysis.

### 4.12. Prediction of the Binding Sites of VNARs on RBD with the ClusPro Server

Protein sequences of VNARs were submitted to the SWISS-MODEL webserver for homology modeling using the default settings. Based on the evaluation results, the highest scoring model was used for molecular docking. RBD was docked using the B chain of PDB ID: 7VOA. The ClusPro website was used for molecular docking of the JM-2, JM-5, JM-17, JM-18, and RBD antigen using the default settings with the docking type of protein. Feedback docking clusters were downloaded from the website for further analysis. Amino acid residues of RBD interacting with VNARs were analyzed using the PDBePISA server. Graphical illustrations were processed using The PyMOL Molecular Graphics System, version 2.5.4 (DeLano Scientific, San Carlos, CA, USA).

### 4.13. Data Analysis

Origin 2019b 32Bit (OriginLab, Northampton, MA, USA) was used for data analysis. The data are expressed as mean ± SD of three replicates. The EC_50_ values were calculated by fitting the OD_450_ values with a sigmoidal dose–response curve.

## 5. Conclusions

In this study, four VNAR binders targeting SARS-CoV-2 RBD were isolated from *C. plagiosum* immunized with RBD. Bi-paratopic VNARs binding with non-overlapping epitope bins were constructed. The monomer VNARs and bi-paratopic VNARs were biochemically characterized. Three bi-paratopic VNARs with RBD-ACE2 blocking ability and high affinity for Omicron and SARS-CoV RBD were identified. These findings highlight the ability and versatility of the diminutive VNAR scaffold for the development of highly specific and effective agents against a given target. Overall, this study provides new insights into screening broad-spectrum antibodies against sarbecovirus.

## Figures and Tables

**Figure 1 ijms-23-10904-f001:**
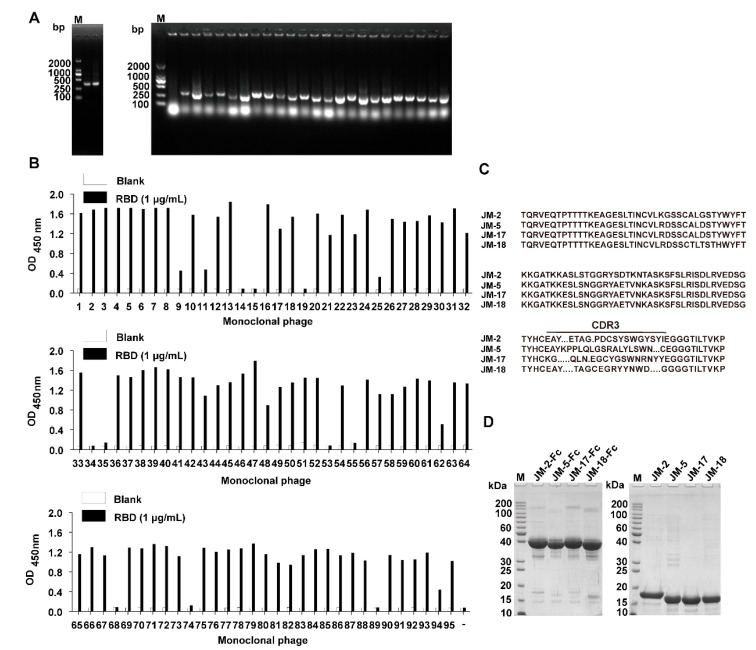
Isolation of anti-SARS-CoV-2 RBD VNARs from RBD-immunized *C. plagiosum* by phage display. (**A**) Amplification of VNAR genes (~340 bp) and determination of the insertion rate of VNAR in the phage library. (**B**) Identification of RBD-specific phages from 95 clones using monoclonal phage ELISA. (**C**) The amino acid sequence of the four identified unique RBD-specific VNARs. (**D**) Reduced SDS-PAGE analysis of the purified VNAR-Fc fusions and Fc-free VNARs.

**Figure 2 ijms-23-10904-f002:**
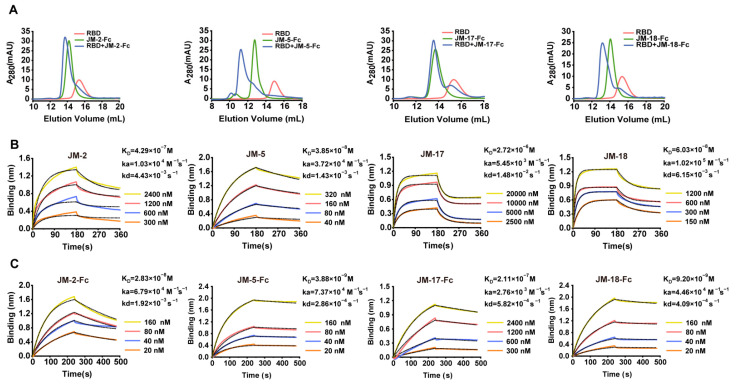
Characterization of the interaction between VNAR fusions and RBD. (**A**) RBD, VNAR-Fc fusions, and their molar mixture (1:1) loaded over a Superdex 200 column. The elution peak from the mixtures appeared earlier than those from RBD and VNAR-Fc fusions. (**B**,**C**) Characterization of binding affinity of isolated VNARs (**B**) or VNAR-Fc fusions (**C**) for RBD using BLI. The actual responses (colored lines) and the data fitted to a 1:1 binding model (black dotted lines) are shown. ka and kd represent the association and dissociation constants, respectively. The equilibrium dissociation constant K_D_ was calculated as kd/ka.

**Figure 3 ijms-23-10904-f003:**
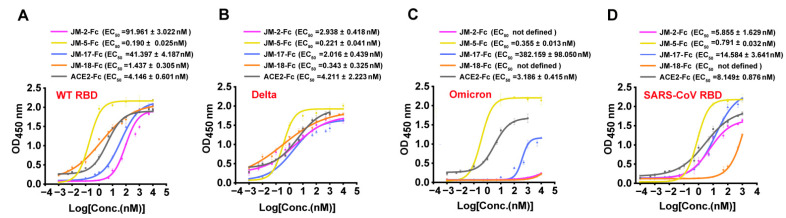
Binding characterization of identified VNARs and ACE2 to various RBD by ELISA. (**A**–**D**) VNAR-Fc fusions binding to RBD of WT (**A**), Delta (**B**), Omicron (**C**), and SARS-CoV (**D**) were characterized using ELISA. Error bars indicate means ± the SD from three independent experiments. The EC_50_ was calculated by fitting the OD_450_ values from serially diluted VNAR-Fc fusions or ACE2-Fc to a sigmoidal dose–response curve.

**Figure 4 ijms-23-10904-f004:**
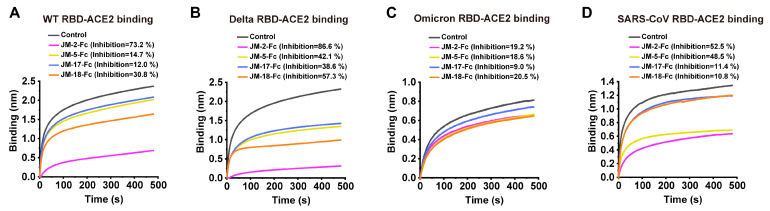
RBD-ACE2 blocking activities of isolated VNARs characterized by BLI assay. (**A**–**D**) Blocking of ACE2 binding to RBD of WT (**A**), Delta (**B**), Omicron (**C**), and SARS-CoV (**D**) by VNARs. Biotinylated RBD was loaded on SA biosensors to a load threshold above 1.0 nm, then incubated with VNAR-Fc fusion and ACE2 sequentially. VNAR was replaced with PBST and used as a control. The inhibition rates of VNARs were calculated against the control group.

**Figure 5 ijms-23-10904-f005:**
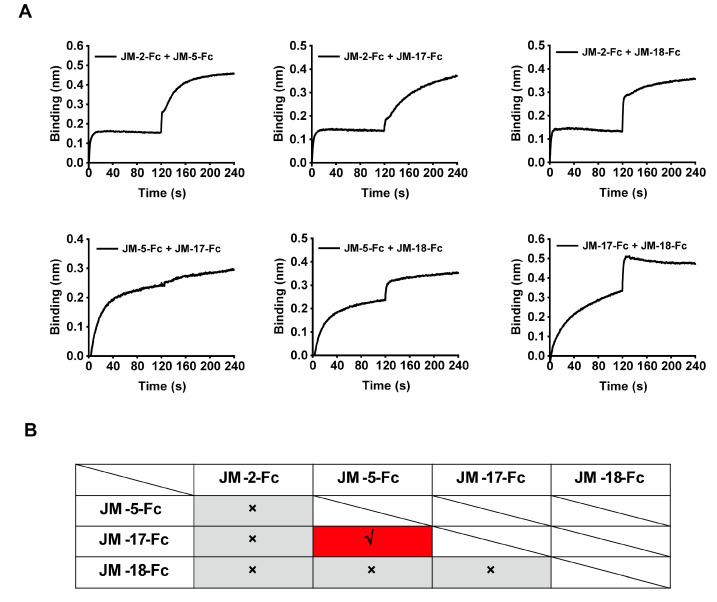
Epitope competition results of the four identified VNAR-Fc fusions by BLI. (**A**) The epitope competition determined by the binding signal transition of two VNAR-Fc fusions. (**B**) The summarized epitope competition results of VNAR-Fc fusions. The red and gray colors indicate competitive and noncompetitive relationships, respectively.

**Figure 6 ijms-23-10904-f006:**
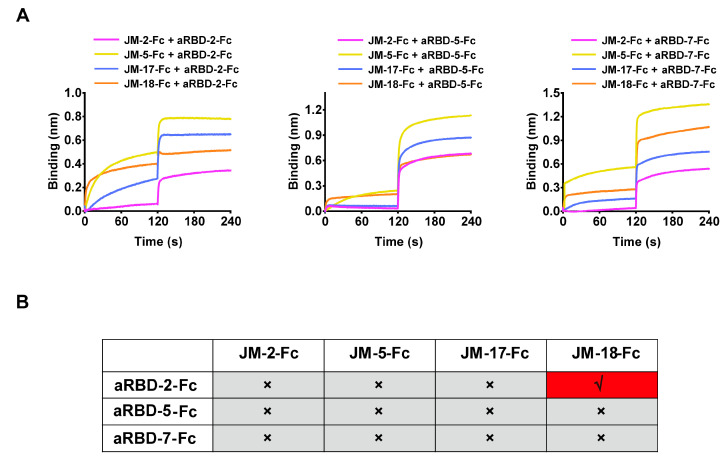
Epitope competition results between the four VNAR-Fc fusions and alpaca-derived VHHs by BLI. (**A**) The competition determined by binding signal transition of shark-derived VNAR-Fc fusions and alpaca-derived VHH-Fc fusions. (**B**) The summarized table for epitope competition of nanobodies from shark and alpaca. The red and gray colors indicate competitive and noncompetitive relationships, respectively.

**Figure 7 ijms-23-10904-f007:**
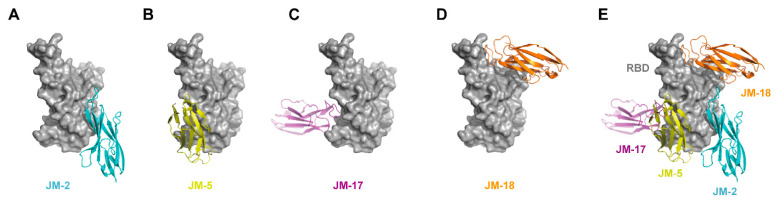
In silico molecular docking of VNARs binding to RBD. (**A**–**D**) Binding mode of JM-2 (**A**), JM-5 (**B**), JM-17 (**C**), and JM-18 (**D**) to RBD. (**E**) The docking structures alignment of JM-2 (cyan), JM-5 (yellow), JM-17 (magenta), and JM-18 (orange) in complex with RBD (gray).

**Figure 8 ijms-23-10904-f008:**
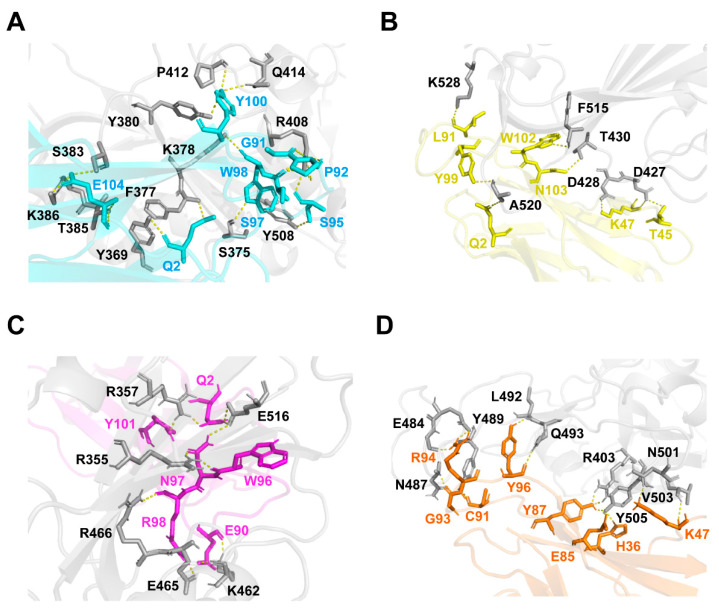
Binding sites of VNARs to WT SARS-CoV-2 RBD. (**A**–**D**) The binding sites of JM-2 (**A**), JM-5 (**B**), JM-17 (**C**), and JM-18 (**D**) to RBD. The binding residues on RBD are shown in gray. The binding residues on JM-2, JM-5, JM-17, and JM-18 are shown in cyan, yellow, magenta, and orange, respectively.

**Figure 9 ijms-23-10904-f009:**
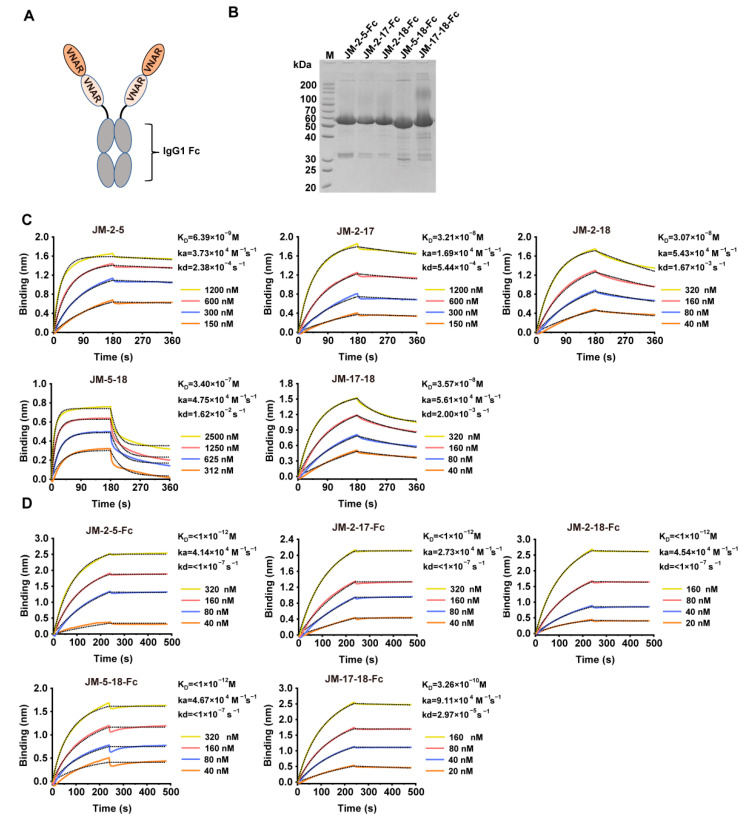
Preparation and characterization of bi-paratopic VNARs. (**A**) Scheme of bi-paratopic VNAR-Fc fusions. (**B**) SDS-PAGE analysis of purified bi-paratopic VNAR-Fc fusions. (**C**,**D**) Binding affinity of the bi-paratopic VNARs (**C**) or VNAR-Fc fusions (**D**) for WT SARS-CoV-2 RBD using BLI. The actual responses (colored lines) and the data fitted to a 1:1 binding model (black dotted lines) are shown. ka and kd represent the association constant and dissociation constant, respectively. The equilibrium dissociation constant K_D_ was calculated as kd/ka.

**Figure 10 ijms-23-10904-f010:**
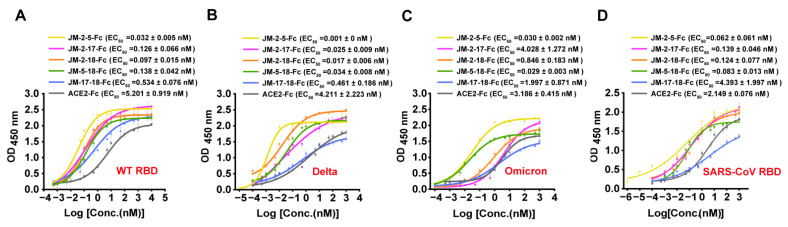
Binding characterization of bi-paratopic VNARs to various RBD by ELISA. (**A**–**D**) The binding of bi-paratopic VNAR-Fc fusions to the RBD of WT (**A**), Delta (**B**), Omicron (**C**), and SARS-CoV (**D**). Error bars indicate means ± SD of three independent experiments. The EC_50_ was calculated by fitting the OD_450_ values from serially diluted VNAR-Fc fusions or ACE2-Fc to a sigmoid dose–response curve.

**Figure 11 ijms-23-10904-f011:**
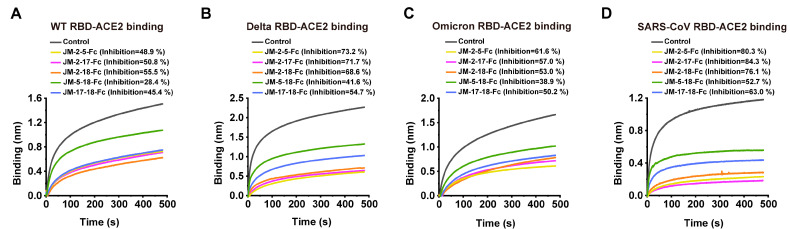
RBD-ACE2 blocking activities of bi-paratopic VNARs characterized by BLI assay. (**A**–**D**) Blocking of ACE2 binding to the RBDs of WT, Delta, Omicron, and SARS-CoV by bi-paratopic VNARs. SA biosensors were loaded with biotinylated RBD, then incubated with bi-paratopic VNAR-Fc fusion and ACE2 sequentially. Bi-paratopic VNAR was replaced with PBST and used as a control. The inhibition rates of bi-paratopic VNARs were calculated against the control group.

**Table 1 ijms-23-10904-t001:** K_D_, ka, and kd values of VNARs, VNAR-Fc fusions, bi-paratopic VNARs, and bi-paratopic VNAR-Fc fusions toward SARS-CoV-2 RBD, and fold of K_D_ increase.

VNAR	K_D_ (M)	ka (M^−1^s^−1^)	kd (s^−1^)	Fold of K_D_ Increase
JM-2	4.29 × 10^−7^	1.03 × 10^4^	4.43 × 10^−3^	N.A.
JM-5	3.85 × 10^−8^	3.72 × 10^4^	1.43 × 10^−3^	N.A.
JM-17	2.72 × 10^−6^	5.45 × 10^4^	1.48 × 10^−2^	N.A.
JM-18	6.03 × 10^−8^	1.02 × 10^5^	6.15 × 10^−3^	N.A.
JM-2-Fc	2.83 × 10^−8^	6.79 × 10^4^	1.92 × 10^−3^	vs. JM-2:15
JM-5-Fc	3.88 × 10^−9^	7.37 × 10^4^	2.86 × 10^−4^	vs. JM-5:10
JM-17-Fc	2.11 × 10^−7^	2.76 × 10^3^	5.82 × 10^−4^	vs. JM-17:13
JM-18-Fc	9.20 × 10^−9^	4.46 × 10^4^	4.09 × 10^−4^	vs. JM-18:6.5
JM-2-5	6.39 × 10^−9^	3.73 × 10^4^	2.38 × 10^−4^	vs. JM-2:149; vs. JM-5:6
JM-2-17	3.21 × 10^−8^	1.69 × 10^4^	5.44 × 10^−4^	vs. JM-2:13; vs. JM-17:86
JM-2-18	3.07 × 10^−8^	5.43 × 10^4^	1.67 × 10^−3^	vs. JM-2:14; vs. JM-18:2
JM-5-18	3.40 × 10^−7^	4.75 × 10^4^	1.62 × 10^−2^	N.A.
JM-17-18	3.57 × 10^−8^	5.61 × 10^4^	2.00 × 10^−3^	vs. JM-17:76; vs. JM-18:1.7
JM-2-5-Fc	<1 × 10^−12^	4.14 × 10^4^	<1 × 10^−7^	>1000
JM-2-17-Fc	<1 × 10^−12^	2.73 × 10^4^	<1 × 10^−7^	>1000
JM-2-18-Fc	<1 × 10^−12^	4.54 × 10^4^	<1 × 10^−7^	>1000
JM-5-18-Fc	<1 × 10^−12^	4.67 × 10^4^	<1 × 10^−7^	>1000
JM-17-18-Fc	3.26 × 10^−10^	9.11 × 10^4^	2.97 × 10^−5^	vs. JM-17-18:109

Note: N.A. means not available.

## Data Availability

Not applicable.

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
