# Peer review of "Screening and Characterization of Shark-Derived VNARs against SARS-CoV-2 Spike RBD Protein"

_ijms, 2022, doi:10.3390/ijms231810904_

Round 1

Reviewer 1 Report

The manuscript titled “Screening and Characterization of Shark-Derived VNARs 2 against SARS-CoV-2 Spike RBD Protein” is focused on screening and characterization of shark derived VNARs. The manuscript is nicely drafted, well organized and relevant for the field. However I found few shortcomings in the present form of the manuscript which I believe needs to be fixed before publication.

Major points:

1. The author did not discuss how glycosylation will impact on binding between VANRs and RBD. This is highly relevant as spike protein is highly glycosylated.

2. What is the binding site of these VNARs to RBD, there is no discussion about that in the manuscript, author should at least try in-sillico docking to determine the binding site, and from the binding site information they can also speculate why there is difference in Kd of different VNARs

3. Complex formation between VNRPs and RBD was only verified by gel-filtration; however this kind of work really needs some sort’s of structural data to confirm the observation and also to study the stoichiometry, however in this manuscript that is completely lacking. What is the stoichiometry of binding between VNRP and RBDs

Minor points:

There are some grammatical mistakes, typos in the manuscript, before final submission authors should thoroughly checked the manuscript.

I also found several formatting error (for e.g. missing superscripts), authors should check them thoroughly.

Author Response

The manuscript titled “Screening and Characterization of Shark-Derived VNARs 2 against SARS-CoV-2 Spike RBD Protein” is focused on screening and characterization of shark derived VNARs. The manuscript is nicely drafted, well organized and relevant for the field. However I found few shortcomings in the present form of the manuscript which I believe needs to be fixed before publication.

Major points:

Q1: The author did not discuss how glycosylation will impact on binding between VANRs and RBD. This is highly relevant as spike protein is highly glycosylated.

A1: We thank the reviewer for raising this issue. Glycosylation of viral proteins is required for the progeny formation and infectivity of SARS-CoV-2. SARS-CoV-2 launches its cellular invasion through its heavily glycosylated S protein. A total of 22 highly occupied N-linked glycosylation sites have been identified in S protein. Further study showed that N-glycosylation of RBD is not only critical for viral internalization but also shields the virus from antibody neutralization. There are two N-linked glycosylation sites (N331 and N343) in SARS-CoV-2 RBD. In this study, the SARS-CoV-2 RBD protein used to immunize sharks was prepared using mammalian cell expression system, N331 and N343 of this RBD protein thus should be glycosylated as done in the natural virus. Therefore, the RBD-targeting IgNARs isolated here should retain binding to the native RBD present on authentic SARS-CoV-2 without being affected by glycosylation. We have add this sentence to the discussion section of revised manuscript. 

Q2: What is the binding site of these VNARs to RBD, there is no discussion about that in the manuscript, author should at least try in-silico docking to determine the binding site, and from the binding site information they can also speculate why there is difference in Kd of different VNARs.

A2: The epitope bins recognized by VNARs were discussed in depth. JM-18 binds to the RBM of RBD and belongs to Class I antibody, while JM-2, JM-5, and JM-17 bind to the RBD core and belong to Class III and Class IV antibody. As kind suggestion from the reviewer, we further performed in-silico molecular docking to predict the binding sites of our VNARs on RBD. The results showed that the ΔG value for JM-2, JM-5, JM-17, and JM-18 were -4.3, -8.3, -2.0, and -3.7 kcal/mol, respectively, which are in line with the KD values of these VNARs. JM-2, JM-5, JM-17 and JM-18 formed 17 hydrogen bonds and two salt bridges, five hydrogen bonds and two salt bridges, ten hydrogen bonds and three salt bridges, and 18 hydrogen bonds and four salt bridges with RBD, respectively. The binding of VNARs to RBD predicted by in-sillico docking was shown in Figure 7 and 8 of the revised manuscript.

Q3: Complex formation between VNRPs and RBD was only verified by gel-filtration; however this kind of work really needs some sort’s of structural data to confirm the observation and also to study the stoichiometry, however in this manuscript that is completely lacking. What is the stoichiometry of binding between VNRP and RBDs.

A3: Actually, in addition to gel-filtration, ELISA and BLI experiments were also conducted to characterize the binding of VNARs to RBD. The binding force between antibody and antigen is called antibody affinity, which is essentially a non-covalent force. It reflects the ability of antibody molecules to bind to the antigen. Antibody affinity is usually expressed by affinity constant KD values, also called equilibrium dissociation constant. BLI is a precise and highly repeatable method for determination of binding KD values of antibody to antigen. In our study, the KD values for JM-2, JM-5, JM-17, and JM-18 to RBD were 429 nM, 38.5 nM, 2720 nM, and 60.3 nM, respectively, while the VNAR-Fc fusions showed enhanced binding affinity, with KD values of 28.3 nM, 3.88 nM, 21.1 nM, and 9.20 nM for JM-2-Fc, JM-5-Fc, JM-17-Fc, and JM-18-Fc, respectively. Numerous studies report SARS-CoV-2 antibodies used BLI to measure the binding affinity of their antibodies (such as [Nat Chem Biol. 2021 Jan;17(1):113-121]; [Nature. 2022 Feb;602(7898):657-663]; [Sci Immunol. 2022 Jul 29;7(73):eabq3511]; and so on). We agree that structural work will further improve our manuscript, and we do believe our nanobodies are still worth reporting, although the mechanism is not clearly revealed yet.

Minor points:

Q4: There are some grammatical mistakes, typos in the manuscript, before final submission authors should thoroughly checked the manuscript. I also found several formatting error (for e.g. missing superscripts), authors should check them thoroughly.

A4: We apologize for this. We have substantially revised our manuscript. 

Besides, to distinguish from previously published RBD-specific antibodies (aRBD-2, aRBD-5), we renamed our VNARs as JM-2, JM-5, JM-17 and JM-18.

Reviewer 2 Report

Due to the ongoing pandemic of coronavirus infection, such works will not lose their relevance for a long time to come. The article is structured and well written.

I recommend it for publication.

Author Response

Due to the ongoing pandemic of coronavirus infection, such works will not lose their relevance for a long time to come. The article is structured and well written.

I recommend it for publication.

A: We thank the reviewer for this kind summary.

Reviewer 3 Report

Screening and characterization of Shark-derived VNARs against SARS_CoV-2 spike RBD protein is a well written and executed manuscript by Yu-Lei Chen et al. However, there are few questions:

1.     Line 101-103, how did you determine the diversity of your library?

2.     Why is the SEC profile of RBD-aRBD-8-Fc is different from other SEC profiles of Fig 2A? Please explain the manuscript.

3.      There has to be inclusion of a reasonable explanation of higher affinity of RBD with VNAR-Fc fusion. Please explain in the manuscript.

4.     Where are the sequences of these VNARs?

Author Response

Screening and characterization of Shark-derived VNARs against SARS_CoV-2 spike RBD protein is a well written and executed manuscript by Yu-Lei Chen et al. However, there are few questions:

Q1: Line 101-103, how did you determine the diversity of your library?

A1: Thanks for your question. To determine the diversity of the library, 100 colonies were selected and subjected to DNA sequencing. The diversity of the VNAR phage library was determined to be 88%.

Q2: Why is the SEC profile of RBD-aRBD-8-Fc is different from other SEC profiles of Fig 2A? Please explain the manuscript.

A2: I wonder if the reviewer was referring to RBD-aRBD-18-Fc instead of RBD-aRBD-8-Fc in our study. To distinguish from previously published RBD-specific antibodies (aRBD-2, aRBD-5), we renamed our VNARs as JM-2, JM-5, JM-17 and JM-18. We used gel filtration chromatography to preliminarily verify the binding abilities of the VNAR-Fc fusions toward SARS-CoV-2 RBD, and found that JM-2-Fc, JM-5-Fc, JM-17-Fc and JM-18-Fc all formed stable complexes with the RBD in solution (Figure 2A). From the SEC profiles, we can conclude that the binding capacity is JM-5-Fc, JM-18-Fc, JM-2-Fc, JM-17-Fc in descending order.

Q3: There has to be inclusion of a reasonable explanation of higher affinity of RBD with VNAR-Fc fusion. Please explain in the manuscript. 

A3: We thank the reviewer for raising this issue. The higher affinity of VNAR-Fc fusions was attributed to bivalent nature of dimerized VNAR-Fc antibody, whose avidity was similar with previously reported nanobodies ([Biochem Biophys Res Commun. 2021 Dec 3;581:38-45.]; [J Virol. 2021 Mar 3;95(10):e02438-20.]; [PLoS One. 2013 Jul 22;8(7):e69495.]; [Biochem Pharmacol. 2018 Dec;158:413-424. ] ).

Q4: Where are the sequences of these VNARs?

A4: The amino acid sequences of VNARs were displayed in Fig 1C.

Reviewer 4 Report

Dear editor,

Find enclose my review for the submitted paper of Yu-Lei Chen et al., “Screening and Characterization of Shark-Derived VNARs 2 Against SARS-CoV-2 Spike RBD Protein». Yu-Lei Chen and coworkers described the selection and characterization of several Shark-derived variable domains of new antigen receptors (VNARs) against SARS-CoV-2 RBD. To that end, C. plagiosum were immunized with the recombinant    SARS-CoV-2 RBD and selection of VNARs was done using phage display. Among the selected VNARs binders, four unique VNARs (aRBD-2, aRBD-5, aRBD-17, and aRBD-18) were obtained, and further characterized. Characterization of each VNAR, in mono or bivalent format has been performed by BLI, ELISA and inhibition experiments. Furthermore, authors have constructed bi-paratopic VNARs and characterized them extensively. The work is suitable for publication in IJMS after minor revisions. Results are clear and well presented.

Minor revisions

(1) Figure 3, the ELISA experiment with the aRBD-5-FC on Delta, omicron and SARS-Cov should repeated with lower concentration as there is no sigmoid curve to calculate the KD.

(2) Even if it’s obvious, Figure 5 and 6, color code should be defined in the figure text (gray or red).

(9) It may be helpful to introduce in Table 1 the improvement of binding for comparison between mono and bivalent (Fold).

(3) I do not understand the meaning of the sentence « the additional steric hindrance caused by the increased size » in the context of avidity effect?

(4) It is better to talk about avidity for the bivalent format.

(5) “BLI” should define also in the main text not only in the materials and methods.

(6) Why in for the better affinity for SARS-CoV than SARS-CoV-2 although the panning as been done on SARS-CoV2 RBD? For example Figure 4 for the aRBD-5-Fc % inhibition 48.5 vs14.7 or itou in Figure 8 for the bi-paratotic aRBD-2-5-Fc.

(7) These results are in correlation with the blocking activities of the bi-paratopic VNARs. Indeed, against why in Figure 9, aRBD-2-5-Fc, aRBD-2-17-Fc and aRBD-2-18-Fc have better blocking activities toward SARS-CoV than SARS-CoV-2? It should be interesting to discuss this issue in the main text.

(8) In addition, for the aRBD-2-5-Fc there is no real sigmoid curve. Lower concentrations should be tested.

Author Response

Dear editor,

Find enclose my review for the submitted paper of Yu-Lei Chen et al., “Screening and Characterization of Shark-Derived VNARs 2 Against SARS-CoV-2 Spike RBD Protein». Yu-Lei Chen and coworkers described the selection and characterization of several Shark-derived variable domains of new antigen receptors (VNARs) against SARS-CoV-2 RBD. To that end, C. plagiosum were immunized with the recombinant SARS-CoV-2 RBD and selection of VNARs was done using phage display. Among the selected VNARs binders, four unique VNARs (aRBD-2, aRBD-5, aRBD-17, and aRBD-18) were obtained, and further characterized. Characterization of each VNAR, in mono or bivalent format has been performed by BLI, ELISA and inhibition experiments. Furthermore, authors have constructed bi-paratopic VNARs and characterized them extensively. The work is suitable for publication in IJMS after minor revisions. Results are clear and well presented.

Minor revisions

Q1: Figure 3, the ELISA experiment with the aRBD-5-Fc on Delta, omicron and SARS-Cov should repeated with lower concentration as there is no sigmoid curve to calculate the KD.

A1: As the suggestion from the reviewer, we repeated the ELISA and updated the results and the Figure 3, please kindly check in the revised manuscript.

Q2: Even if it’s obvious, Figure 5 and 6, color code should be defined in the figure text (gray or red).

A2: The color code has been defined in legends of Figure 5 and 6 in the revised manuscript.

Q3: I do not understand the meaning of the sentence « the additional steric hindrance caused by the increased size » in the context of avidity effect?

A3: We apologize for this incorrect formulation, we have remove this sentence from the revised manuscript.

Q4: It is better to talk about avidity for the bivalent format.

A4: Thanks for your question. The avidity of bivalent VNARs have been discussed in the last paragraph of the Discussion section.

Q5: “BLI” should define also in the main text not only in the materials and methods.

A5: As suggested, “BLI” has been defined in the Results section of the revised manuscript. 

Q6: Why in for the better affinity for SARS-CoV than SARS-CoV-2 although the panning as been done on SARS-CoV2 RBD? For example Figure 4 for the aRBD-5-Fc % inhibition 48.5 vs14.7 or itou in Figure 8 for the bi-paratotic aRBD-2-5-Fc. These results are in correlation with the blocking activities of the bi-paratopic VNARs. Indeed, against why in Figure 9, aRBD-2-5-Fc, aRBD-2-17-Fc and aRBD-2-18-Fc have better blocking activities toward SARS-CoV than SARS-CoV-2? It should be interesting to discuss this issue in the main text.

A6: To distinguish from previously published RBD-specific antibodies (aRBD-2, aRBD-5), we renamed our VNARs from aRBD-2, aRBD-5, aRBD-17 and aRBD-18 to JM-2, JM-5, JM-17 and JM-18 correspondingly.

aRBD-2, aRBD-5 and aRBD-7 are previously descripted nanobodies derived from alpacas, our previous study has determined the structures of these nanobodies in complex with RBD and showed that aRBD-2 binds to the lateral loop of receptor-binding motif (RBM) of RBD, while aRBD-5 and aRBD-7 bind to the concave surface anchored by the β-hairpin of RBM. These three nanobodies collectively occupy most surface region of the RBM. Therefore, antibodies that do not compete with these three nanobodies for RBD binding should be located on outside of RBM, that is, the core region. In this study, shark-derived JM-18 was shown to clash with aRBD-2 for binding RBD, while JM-2, JM-5 and JM-17 were shown have no competition with aRBD-2, aRBD-5 and aRBD-7 in binding RBD, suggesting that JM-18 may bind to the RBM, while JM-2, JM-5 and JM-17 may target core. Antibodies targeting the RBD core often have SARS-CoV-1 and SARS-CoV-2 cross-reactivity, as the core is relatively conserved in SARS-CoV-1 and SARS-CoV-2. Herein, JM-2 showed broad RBD-ACE2 blocking activities against WT and Delta variant of SARS-CoV-2 as well as SARS-CoV. Although these VNARs was screened against SARS-CoV-2 RBD, JM-5 exhibited higher blocking capacity against SARS-CoV RBD than SARS-CoV-2 RBD, indicating that the binding of JM-5 toward SARS-CoV RBD might be much stronger. Therefore, further studies should determine the KD values of these VNARs toward SARS-CoV RBD by BLI and explore the binding sites of these VNARs toward RBD by crystallization. 

Interestingly, JM-2 fused bi-paratopic VNARs (JM-2-5-Fc, JM-2-17-Fc, and JM-2-18-Fc) showed potent ACE2-RBD blocking ability in SARS-CoV-2 variants and SARS-CoV, further illustrating that JM-2 recognized a conserved epitope across sarbecovirus clades. These results provide new insights into screening broad-spectrum antibodies against sarbecovirus.

Q7: In addition, for the aRBD-2-5-Fc there is no real sigmoid curve. Lower concentrations should be tested.

A7: Thanks for reminding. We have repeated the experiment and update the results in revised Figure 10.

Q8: It may be helpful to introduce in Table 1 the improvement of binding for comparison between mono and bivalent (Fold).

A8: As suggested, we have added the fold information in Table 1.

Round 2

Reviewer 1 Report

I am convinced with the revised version of the paper, So I recommend it for publication